Catalases in the pathogenesis of Sporothrix schenckii research

Vargas-Maya Naurú Idalia 1
Olmedo-Monfil Vianey 1
Ramírez-Prado Jorge Humberto 2
Reyes-Cortés Ruth 1
Padilla-Vaca Felipe padillaf@ugto.mx 1
Franco Bernardo bfranco@ugto.mx 1
1 Biology Department, Universidad de Guanajuato , Guanajuato , Guanajuato , México
2 Unidad de Biotecnología, Centro de Investigación Científica de Yucatán , Merida , Yucatán , México
Sistla Srinivas
Electronic publication date: 2022 Dec 7
Publication date: 2022
Volume: 10
Electronic Location ID: e14478
Received 2022 Sep 8; Accepted 2022 Nov 7
Copyright: ©2022 Vargas-Maya et al.
Copyright year: 2022
Copyright holder: Vargas-Maya et al.
License: This is an open access article distributed under the terms of the Creative Commons Attribution License, which permits unrestricted use, distribution, reproduction and adaptation in any medium and for any purpose provided that it is properly attributed. For attribution, the original author(s), title, publication source (PeerJ) and either DOI or URL of the article must be cited.
License URL: https://creativecommons.org/licenses/by/4.0/

Keywords: Fungal catalase, Virulence determinant, Sporothrix schenckii, Reactive oxygen species, Innate immunity

Funding: The authors received no funding for this work.

==============================
Pathogenic fungal infection success depends on the ability to escape the immune response. Most strategies for fungal infection control are focused on the inhibition of virulence factors and increasing the effectiveness of antifungal drugs. Nevertheless, little attention has been focused on their physiological resistance to the host immune system. Hints may be found in pathogenic fungi that also inhabit the soil. In nature, the saprophyte lifestyle of fungi is also associated with predators that can induce oxidative stress upon cell damage. The natural sources of nutrients for fungi are linked to cellulose degradation, which in turn generates reactive oxygen species (ROS). Overall, the antioxidant arsenal needed to thrive both in free-living and pathogenic lifestyles in fungi is fundamental for success. In this review, we present recent findings regarding catalases and oxidative stress in fungi and how these can be in close relationship with pathogenesis. Additionally, special focus is placed on catalases of Sporothrix schenckii as a pathogenic model with a dual lifestyle. It is assumed that catalase expression is activated upon exposure to H2O2, but there are reports where this is not always the case. Additionally, it may be relevant to consider the role of catalases in S. schenckii survival in the saprophytic lifestyle and why their study can assess their involvement in the survival and therefore, in the virulence phenotype of different species of Sporothrix and when each of the three catalases are required. Also, studying antioxidant mechanisms in other isolates of pathogenic and free-living fungi may be linked to the virulence phenotype and be potential therapeutic and diagnostic targets. Thus, the rationale for this review to place focus on fungal catalases and their role in pathogenesis in addition to counteracting the effect of immune system reactive oxygen species. Fungi that thrive in soil and have mammal hosts could shed light on the importance of these enzymes in the two types of lifestyles. We look forward to encouraging more research in a myriad of areas on catalase biology with a focus on basic and applied objectives and placing these enzymes as virulence determinants.

Introduction

When cells are exposed to oxidative stress, specifically H2O2, it is assumed that antioxidant enzymes are induced and perform their task to detoxify the cell milieu. However, this is not always the case; sometimes, antioxidant enzymes are damaged by the same molecules they should eliminate (Karakus, 2020; Nicholls, 2012).

Vertebrates use hydrogen peroxide as a biological weapon in combination with other molecules to potentiate its effect. This is particularly efficient for damaging the pathogen’s DNA (Mahaseth & Kuzminov, 2017), resulting in a more complicated task to survive the immune response.

Pathogens encode various antioxidant molecules, including catalases. Catalases (EC 1.11.1.6) are heme-containing enzymes that catalyze the dismutation of hydrogen peroxide (2H2O2) into 2H2O and oxygen (O2) (reaction 1). The catalytic reaction steps are as follows: (1) 2H2O2→2H2O+O2

(2) EnzPor−FeIII+H2O2→CpdIPor+•−FeIV=O+H2O

(3) Cpd IPor+•−FeIV=O+H2O→EnzPor−FeIII+H2O+O2

(4) Cpd IPor+•−FeIV=O+AH2→Cpd IIPor−FeIV−OH+AH•

(5) Cpd IIPor−FeIV−OH+H2O2→CpdIIIPor−FeIII−O2−•+H2O.

The first step is the oxidation of the heme using first hydrogen peroxide molecule to form an oxyferryl species resulting in a porphyrin cation radical (reaction 2, compound I). This compound I is reduced by a second hydrogen peroxide to regenerate the resting enzyme state, producing water and oxygen (reaction 3). Catalases can also have peroxidase activity with suitable organic compounds (transition from compound I to II in reaction 4). Compound II can be oxidized by another hydrogen peroxide resulting in the inactive compound III in reaction 5 (Karakus, 2020).

Catalases are widespread in aerobic organisms and have been linked to survival during oxidative stress (Karakus, 2020; Nicholls, 2012). Catalases are homotetrameric proteins containing a heme group buried deep in the protein. The access to the catalytic domain is through a 45 Å channel where H2O2 residence is enhanced, rendering a selectivity for this substrate (Domínguez, Sosa-Peinado & Hansberg, 2014) and having evolved to exclude water molecules by displacing water molecules embedded in the active site using Phe170, Phe171 and Phe178 and the role of the negative charge from Asp145; this allows a high kinetic activity (which the km is in the range of 20 to 200 mM) (Domínguez, Sosa-Peinado & Hansberg, 2010; Hansberg, Salas-Lizana & Domínguez, 2012).

The sequence and structure of catalase domains are more divergent than previously thought. This feature has rendered the classifications of these enzymes in three clades (Domínguez, Sosa-Peinado & Hansberg, 2010; Horvath & Grishin, 2001). Clade I refers to catalases from plants, green algae, and Clade III to archaea, bacteria, fungi, and animals (Domínguez, Sosa-Peinado & Hansberg, 2010). These clades are proteins with subunits of 55 to 69 kDa. Clade II belongs to bacteria, archaea, and fungi and is formed by larger subunits of 75 to 86 kDa; the additional residues are located in the C-terminal domain and belong to type 1 glutamine amidotransferase (Horvath & Grishin, 2001).

Catalases have complex reaction mechanisms for a simple dismutation reaction, which has been a hot research topic. Although much information is available, it mostly focuses on bacteria and some examples of fungal catalases. Nevertheless, catalases are still being studied due to their diversity among prokaryotic and eukaryotic organisms. One example is a catalase found with phenol oxidase activities and the interchange of activities between catalase and phenol oxidase in the fungus Scytalidium thermophilum (Sutay Kocabas et al., 2008). This has been observed to be relevant in polyphenol oxidation, where H2O2 is released (Akagawa, Shigemitsu & Suyama, 2003), thus affecting the free-living lifestyle of bacteria and fungi. These enzymes have been demonstrated to have a bacterial origin (Bacteroidetes) and have been found in another Ascomycota (Kamlárová, Chovanová & Zámocký, 2018).

In the case of some parasites that do have catalases, these enzymes have been demonstrated to play a key role against host defense mechanisms and survival. In some cases, only one catalase gene is present, but an important arsenal of other Reactive Oxygen Species (ROS) detoxifying enzymes are needed for survival (Kwok et al., 2004; Staerck et al., 2017), adding to our current understanding of the pathogenesis of protists.

In the literature, there are experimental conditions where fungal catalases are induced and needed for survival such as temperature shift to 37 °C in C. neoformans, with a focus on the signal transduction pathways, such as MAPK or phosphorelay pathways resulting in the activation of the AP-I family of transcription factors that regulate their expression (Aguirre, Hansberg & Navarro, 2006). Nevertheless, in fungal pathogens, this is not fully addressed because the best-studied Ascomycete catalases are encoded in the genome of Neurospora crassa, which have a link between morphogenesis and cell differentiation as well as for contending with environmental stressors (Aguirre et al., 2005; Fountain et al., 2016). Additionally, extensive structural studies have been carried out on N. crassa catalases, showing unique features for H2O2 binding and recognition in a water milieu (Domínguez, Sosa-Peinado & Hansberg, 2010) and complex inhibitory mechanisms by singlet oxygen (O=O) reducing its stability and resistance to degradation (Díaz et al., 2005). In the case of bovines, catalase possesses resistance to singlet oxygen, the dismutation of hydrogen peroxide occurs without generating oxygen (de Groot et al., 2006). In turn, this endurance to O=O is not known in pathogenic fungi and may become a potential target for treatment using other inhibitors (Kim, Kwon & Park, 2001).

In Candida albicans, the high expression of these enzymes may result in reduced fitness. High expression levels in clinical isolates result in a double-edged sword; on the one hand, it protects cells from oxidative stress conditions, but on the other hand, in the absence of stress, it reduces cell fitness by the increase in iron demand, thus this is alleviated by iron supplementation. Therefore, the reduction in fitness is less likely to happen in iron rich environments such as the kidney or spleen in a mouse model, suggesting that pathogen colonization is linked to catalase expression (Pradhan et al., 2017).

ROS production in fungal organisms varies with metabolic states and cell damage; and asexual development is closely related to ROS present in the environment. When catalases are absent, the asexual cycle of the cell differentiation program is enhanced in N. crassa (Michán, Lledías & Hansberg, 2003; Zamocky, Furtmüller & Obinger, 2009). Catalase expression, for instance, is related to redox balance control in fungal plant pathogens, such as Sclerotinia sclerotiorum, where this enzyme is needed for cell integrity, oxidative stress resistance, pathogenicity, and antifungal resistance (Huang et al., 2021). What is truly striking in S. sclerotiorum is that the genome encodes seven catalases. Nevertheless, only one contributes to oxidative stress resistance (Huang et al., 2021). The role of the other catalases and their regulation remains to be explored.

Determining the importance of catalases may impede the discovery of novel potential uses in diagnosing and treating pathogenic fungi. One such example is the presence of circulating antibodies in patients infected with Histoplasma capsulatum that recognize catalases B, M antigen, and P, serving as potential targets for diagnosis kits (Almeida et al., 2020), and these enzymes have been demonstrated to be required for virulence (Holbrook et al., 2013; Johnson et al., 2002).

For all the above, this review addresses the following question: why have catalases been neglected in pathogenic fungi research as both potential targets for treatment and diagnosis? One important aspect that partially explains this is that these enzymes are assumed to be highly conserved and functionally defined in all kingdoms of life. However, oxidative stress has different outcomes in distinct organisms. Likewise, this review proposes a closer look on Sporothrix schenkii as an example of an emerging fungal pathogen with an evolutionary well-adapted saprophytic lifestyle.

The Sporothrix pathogenic clade is considered a neglected tropical and subtropical disease since the incidence is not mandatory for health authorities to notify (Gremião et al., 2021). The disease, usually caused by S. schenckii, S. brasiliensis, S. globosa and S. lureiei, is characterized of cutaneous and subcutaneous disease that rarely affects deep-seated organs (Mora-Montes, 2022), and the best-studied structure of these organisms is the cell wall (Mora-Montes, 2022). But other aspects of its physiology and virulence determinants are at their onset. Here, we propose that catalases may be key players in cell survival, resulting in better colonization of the host and thus resulting in local o disseminated disease, which may be related to the well-adapted physiology of the saprophytic lifestyle of this genus.

Methodology

The literature was consulted through Pubmed and Google Scholar. Key words used were ‘Catalase’, ‘Pathogenic fungi’, ‘Sporothrix schenckii’, and the Boolean ‘and’ for the combination of these keywords. Authors conducted independent review of the literature to prevent any bias, and the selected articles were chosen as recent as possible. When selecting the studies to be included in this review, the number of articles addressing the role of catalases in pathogenic fungi is scant. Here, we aimed to provide as much information as possible with the available literature. In Table S1, the articles with the most relevant topics for this review are briefly summarized in alphabetic order.

The sequence analysis was conducted using BLASTp (Altschul et al., 1990). Protein structure prediction was conducted using AlphaFold2 (Jumper et al., 2021) with the default options, using the API hosted at Söding lab based on MMseqs2 server (Mirdita, Steinegger & Söding, 2019). Dimer prediction of the three catalases of S. schenckii was performed with AlphaFold Multimer prediction suite using the default parameters (Evans et al., 2021) Phylogenetic analysis was conducted with MEGA version 11.0.13 (Tamura, Stecher & Kumar, 2021). In brief, the evolutionary history was inferred by using the Maximum Likelihood method and JTT matrix-based model (Jones, Taylor & Thornton, 1992) using protein sequences aligned with ClustalW in MEGA. The tree with the highest log likelihood (−10,615.68) is shown. The percentage of trees in which the associated taxa clustered together is shown below the branches. Initial tree(s) for the heuristic search were obtained by applying the Neighbor-Joining method to a matrix of pairwise distances estimated using the JTT model and 500 bootstrap. This analysis involved seven amino acid sequences. There was a total of 844 positions in the final dataset. KatG from Escherichia coli was used as an outgroup (accession number P13029, Uniprot). Active site sequence comparison was achieved by aligning the three N. crassa and S. schenckii catalase sequences with ClustalW with the default settings, then exported to Weblogo 3 (Crooks et al., 2004). Manually, the catalytic residues were indicated.

Protein structure alignment was conducted with mTM-aling (Dong et al., 2018) using the default settings. Protein structures included the AlphaFold2 models of S. schenckii and the PDB files of N. crassa catalases.

The Case of Pathogenic Fungi: Sporothrix Schenckii

In the genome sequence of S. schenckii, three catalase coding genes were identified based on homology to Aspergillus and Neurospora genes. In RedoxiBase (http://peroxibase.toulouse.inra.fr/) (Savelli et al., 2019), only one catalase is annotated for S. schenckii (as KatE, accession number XP_016592737.1 or SPSK1099_11725-RA in the S. schenckii genome database). However, at least three were identified by BLAST analysis and expressed in response to oxidative stress (Román-Casiano et al., 2021). The work by Román-Casiano et al. (2021) described the response of these three catalases in the presence of H2O2 and the relative expression levels, showing that Cat1 (ERS99939.1), one of the small catalases, is highly expressed and resulted in the predominant activity upon H2O2 exposure. The second catalase that is highly expressed is the large subunit catalase (81.4 kDa, accession number ERT00986.1), while a third catalase showed low activity. When analyzing several fungi in RedoxiBase, the repertoire found for antioxidant enzymes is vast and varied in all species; this imposes a challenge when assessing their role, specifically in cases where two contrasting lifestyles are found in the same organism. In Ascomycota alone, catalases and catalase/peroxidases are the fourth most abundant antioxidant enzymes. The three front runners ahead of catalases are cytochrome C peroxidase, fungi-bacteria glutathione peroxidase, and hybrid ascorbate-cytochrome C peroxidase.

In the work by Román-Casiano and colleagues (2021), two isoforms (CAT1 and CAT 3, accession numbers: ERS99939.1 and ERT00986.1, respectively) were shown to be highly expressed upon exposure to oxidative stress. However, in a recent paper, Saucedo-Campa and collaborators showed that this organism’s landscape is more complex than previously thought. Several moonlight proteins (Hsp70-5, lipase 1, enolase, and pyruvate kinase, for example) are induced by oxidative stress by H2O2 (Saucedo-Campa et al., 2022), suggesting that the arsenal for H2O2 detoxification in this organism is complex and involves proteins previously thought to be related to protein folding, lipid metabolism, or even metabolic enzymes that in the cell wall may represent the first line of defense. Additionally, in the case of menadione-induced oxidative stress, other moonlight proteins (for example, β-1,3-endoglucanase, glycoside hydrolase, chitinase, Hsp30, lipase, trehalase) are present in the cell wall as protection against oxidative stress (Félix-Contreras et al., 2020). Lipase seems to be induced in two distinct oxidative stress conditions; further research is needed to assess the contribution of this and other moonlight proteins present in the cell wall that may have additional antioxidant roles in S. schenckii.

In the case of the catalases of S. schenckii, structural features can now be modeled with accuracy. The sequence features of the three catalases encoded in the S. schenkii genome suggest that these enzymes may play different roles depending on the organism’s morphological state as either free-living or as a pathogen. In Fig. 1A, BLAST analysis shows that the main homologs of S. schenckii catalases are clustered (Figs. 1B and 1C), indicating that Cat2 is the most divergent catalase in this comparison. The variation in catalytic residues poses the question of whether the catalases of S. schenckii have different kinetic parameters and may respond differently to oxidant agents and other molecules present in the media (see below).

Figure 1 Sequence and structural features of Sschenckii catalases.

(A), BLAST analysis was used to identify the closest homologs for the three catalases of S. schenckii, and 100 hits were downloaded and visually represented in pairwise identity 2D maps with Alignment Viewer (https://alignmentviewer.org/). In (A), pairwise identity 2D maps are shown for the three catalases. The number of hits for catalase 1 (ERS99939.1) was 132. For catalase 2 (ERS95255.1), 177 hits were obtained, and for catalase 3 (ERT00986.1), 140 hits were obtained. Catalase 2 shows lower homology with the cognate orthologs than catalase 1 or 3. In (B), Phylogenetic analysis of the three catalases from Neurospora crassa and S. shenckii (Phylogenetic analysis was conducted with MEGA version 11.0.13 (Tamura, Stecher & Kumar, 2021), KatG (Uniprot P13029) was used as outergroup. In (C), Weblogo fragments representing the regions with the active site residues from the sequence alignment between N. crassa and S. schneckii catalases. Red arrows indicate conserved catalytic residues in all sequences, and blue arrows represent residues identified from the catalytic core but are not conserved in all catalases (data retrieved from Díaz et al. (2009)).

Figure 2 Conserved structural features of S. schenckii catalases compared with N. crassa experimentally determined structures.

(A), protein dimers are represented as ribbon and rainbow of N. crassa catalases. The PDB number is indicated. 1SY7 is the large subunit catalase/peroxidase, and 5WHS and 4BIM are the small subunit catalases. Relevant domains are indicated in the large subunit catalase, and heme is indicated with white arrows. (B), AlphaFold2 models of the S. schenckii catalases, indicating the N and C terminal ends. Asterisk suggest putative heme site. (C), structural alignment with the three N. crassa catalases (RMSD 1.15). Reference structures are indicated, in blue is PDB 1SY7, in green is PDB 4BIM, in red is PDB 5WHS, in yellow is catalase 1 (ERS99939.1), in light blue is catalase 2 (ERS95255.1), and in purple is catalase 3 (ERT00986.1). Conserved residues are indicated in magenta. Structural alignment was conducted with mTM- align (Dong et al., 2018).

The other aspect to consider with catalases is the conservation of structural features. In Fig. 2A, shows the previously high-resolution crystal structure reported for N. crassa catalases, which have been studied in detail (Díaz et al., 2009). Future research can be focused on structural comparisons with other fungal organisms and may ultimately lead to the study of the kinetic and structural features of other fungal catalases. As shown here, Cat1 and Cat2 of S. schenckii are small catalases, while Cat3 is a member of the large catalases.

In the case of catalase 1, the relevant BLAST hits are with catalases from Ascomycetes such as Ophiostoma piceae, Diaporthe sp., Valsa mali, Hypoxylon sp., among other plant pathogens (Fig. 1A). Here, the phylogenetic distribution is wider than that observed for the other two catalases. This is shown in Fig. 1B, where S. schenckii catalases are compared with the three best-matching homologs of different species, showing homology to catalases from plant pathogens or plant-associated fungi. This correlates with the saprophytic lifestyle of S. schenckii and perhaps catalase 3 is more restricted to survival inside the host rather than withstanding the environmental conditions in the saprophytic stage.

For catalase 2, the homology with BLAST hits is the lowest of the three catalases, and the highest proteins showing homology are derived from Fusarium, Trichoderma, Aspergillus, and Penicillium species. However, the homology found is lower than that observed with the other two catalases (Fig. 2B).

Regarding catalase 3, we found homology to catalases from ascomycete fungi such as Coniochaeta sp, Thozetella sp, Podospora anserina, and others with similar lifestyles, and is strikingly similar to Catalase 1 from N. crassa. The most distant hit is with the bioluminescent basidiomycete Mycena chlorophos. Overall, this is consistent with the previous report of Román-Cansiano on identifying these enzymes and renders a potential specific role of each catalase while growing in a saprophytic stage or during the interaction with the host (Román-Casiano et al., 2021).

One interesting feature of these S. schenckii catalases is that the catalytic residues are not conserved, especially the catalytic triad Arg 87 (conserved), tryptophan 90 (not conserved, replaced by valine), and histidine 91 (conserved) (Zamocky, Furtmüller & Obinger, 2009; Díaz et al., 2009) (Fig. 1C indicated with a red rectangle), which may have contrasting affinities for H2O2 or inhibitory molecules (Karakus, 2020).

In S. schenckii, the expression patterns of the catalase genes in transcriptomic data (Giosa et al., 2020) and http://sporothrixgenomedatabase.unime.it are as follows: the highest expressing enzyme in the yeast form is Cat 3 ( ERT00986.1) at 7.38 log2FC. For Cat1 (ERS99939.1), it is 5.44 log2FC in the yeast form. Finally, Cat 2 (ERS95255.1) was not found in the transcriptome analysis between morphologies, consistent with the findings by Román-Casiano and colleagues (2021), where even in the presence of H2O2, its expression is low. However, the zymogram analysis using exponentially growing yeast cells shown by Román-Casiano et al. (2021) suggests that the three catalases are expressed, and in high H2O2 concentrations, Cat3 loses its activity completely, and a decrease in overall catalase activity is observed. This may impact the infection progression by limiting or blocking the growth of the microorganism.

Overall, the catalase-encoding gene distribution is complex. Even with extensive genomic data, these enzymes’ congruent analysis and evolutionary aspects have been carried out in fungi, especially in pathogenic fungi (Passardi et al., 2007). Biochemical data on these enzymes are also missing, particularly regarding H2O2 affinity, catalytic rate, and inhibitors.

The structure of fungal catalases shows that the large and small subunit catalases contain well-defined domains (Fig. 2A). The heme is deeply buried in the active site and is accessible via a 45 Å  tunnel. Close inspection of the catalase models from S. shcenckii suggests that small subunit catalases are more structurally divergent from N. crassa homologs. Overall, the conserved residues are in the vicinity of the active site. Cat3 from S. schenckii shows a conserved structure compared to the well-defined N. crassa large subunit catalase (Fig. 2). AlphaFold multimer prediction rendered that the three catalases of S. schenckii form dimers (Fig. 3). The dimers showed one interesting feature, the N-terminal end is not embedded in the structure as has been shown in N. crassa catalases. This suggests that different stability in the protein to denaturing agents or inhibitory molecules may characterize these catalases. Also, additional, or other residues implicated in excluding water from the active site, the different effects of inhibitors, and different kinetic parameters may be exclusive to these catalases. They may be relevant in the two environmental conditions S. schenckii thrives. Further biochemical studies will clarify if these molecules can be targets of inhibitors that may result in better management of infected hosts.

Figure 3 S. schenckii catalases are predicted to form dimers similar to N. crassa catalases.

Upper panels are reference catalases as shown in Fig. 2. Lower panels are the AlphaFold multimer predictions. Each chain is indicated with a different color, N and C terminal ends are indicated along with the putative heme-binding site (asterisk).

Further analysis of the cumulative genomic data may shed light on the sequence and structural differences of catalases related to differences in catalysis and stability, subcellular localization, and turnover. A surprising role for catalases was found by Nava-Ramírez & Hansberg (2020), who demonstrated that the C-terminal domain of the large-size subunit catalase from N. crassa possesses chaperone activity that is absent in small subunit catalases. When this C-terminal domain is transferred to small subunit catalases, it functions as a chaperone as well, rendering a more stable enzyme not only for H2O2 but also for other stress conditions (Hansberg et al., 2022). C-terminal domain originated from the fusion of the bacterial small subunit catalase and Hsp31 chaperone (Hansberg et al., 2022). The chaperone activity is closely related to the effect of ROS and the misfolding of proteins, rendering catalases a secondary tool for preventing cell damage. The structural features found in catalase 3 of S. schenckii may also possess this activity (Fig. 2B, catalase 3), which is also relevant during exposure to innate immune cells, due to the high production of reactive oxygen and nitrogen species which damage proteins.

The biochemical features of S. schenckii catalases and experimental determination of their structure are lacking. Additionally, their role in infection has not been studied in detail. The evidence suggests that these enzymes are relevant to oxidative stress, but further research is needed.

The next step for assessing the role of the antioxidant response in the Sporothrix complex since S. schenkii and S. brasiliensis possess different resistance to hydrogen peroxide and menadione, being the latter more resistant in the MYA 4843 strain (Ortega et al., 2015). This supports the need to assess the regulation and specific differences in all antioxidant-regulatory and effector proteins in all Sporothrix species to assess their relevance in different virulence phenotypes.

The finding by Ortega and colleagues (Ortega et al., 2015) that in some instances in S. schenckii and S. brasiliensis there are more than one antioxidant enzyme suggests not redundancy, but specific roles of these enzymes, along with the complex regulatory network that has been elucidated in other fungi (Aguirre, Hansberg & Navarro, 2006). The components of the signal transduction pathway leading to the regulation of antioxidant enzymes, there are putative proteins involved in the process with low homology to bona fide regulatory proteins from Saccharomyces cerevisiae and Candida albicans (Ortega et al., 2015), suggesting perhaps a more diverse role in the Sporothrix genus than previously thought. The antioxidant arsenal has been demonstrated to be essential for the colonization since in experimentally infected rats, the infection by S. schenckii causes an extensive inflammatory response with a rise in general oxidative state and worsening the outcome of the infection and aggravating the clinical condition of the host, resulting in a strong redox imbalance that ultimately affects host and pathogen alike (Castro et al., 2017). Further research regarding both the redox balance in the host and the complete regulatory pathway may contribute to deepening the understanding of the Sporothrix genus pathogenesis.

Overall, the major limitations of lacking profound knowledge of the antioxidant mechanisms, specifically of catalases, are the following: are each isoform of catalases specific to a cell morphology or differentiation stage? Is the regulation of each catalase the same? the structural features of the catalases in the Sporotrhix genus provide different catalytic mechanisms? Are these catalases sensitive to novel inhibitors? One major issue is the difficulty of generating deletion mutants in the Sporothrix genus. Thus, assessing the role of single and multiple mutants of catalases poses a major challenge for in vivo analysis.

Future Research

The study of both the free-living and the pathogenic lifestyle of S. schenckii and other species of the Sporothrix genus is relevant to understanding dissemination and zoonosis. In the case of fungi that interact with plant hosts, such as Trichoderma atroviride, its genome encodes two catalase-peroxidases (http://peroxibase.toulouse.inra.fr/). For T. atroviride, the role of these catalases has not been addressed, but KatG2 (TatKatG2) sequence analysis suggests that it is a secreted enzyme (Zamocky, Furtmüller & Obinger, 2009).

An important feature of oxidative stress and radical detoxifying enzymes is linked to cell damage in T. atroviride. Hernández-Oñate and colleagues (2012) described that NADPH oxidase-dependent ROS production is linked to development upon physical cell injury. H2O2 and oxylipins are signaling molecules shared in all kingdoms of life that respond to oxidative damage. Moreover, catalase 2 is downregulated in transcriptomic data, suggesting that H2O2 isa part of the signaling for injury repair and needs to accumulate in the hyphae; this remains an open question in the case of pathogenic fungi and the role of ROS in the differentiation process, cell damage and the regulation of cell death mechanisms (Hernández-Oñate et al., 2012). If plant-associated fungi, there is a particular role for a catalase in cell damage, we hypothesize that perhaps one of the three catalases in S. schenckii may be involved in the saprophytic lifestyle and not so needed during host colonization.

Oxidative stress is linked to cellulose degradation and involves the generation of hydroxyl radicals via the Fenton reaction from the H2O2 produced by the lytic polysaccharide monooxygenases (LPMOs) secreted by fungi (Li et al., 2021; Castaño et al., 2018). ROS that are produced in this process also have a deleterious effect on antioxidant enzymes such as oxidases, glutathione S-transferases, and thioredoxins, which may increase cell damage by reducing antioxidant enzymes (Castaño et al., 2021), while glycoside hydrolases are adapted to operate in such conditions. Taking the data from Román-Casiano et al. (2021) and the observation that cellulose degradation requires and exacerbates ROS production and antioxidant enzymes are sensitive to this environmental insult, it is tempting to test catalase activities in Sporothrix and other pathogenic fungi growing with cellulose as a carbon source and to test which catalase is more active or is resistant to oxidative stress during the free-living lifestyle of these organisms. For instance, it remains to be analyzed whether the expression of a β-glucosidase with transglycosylation and cellulase activities are involved in the in vivo cellulolytic complex of S. shenckii saprophytic lifestyle (Hernández-Guzmán et al., 2016).

The regulatory pathways for the antioxidant response are also diverse in fungi. The antioxidant counteracting transcription factors are also involved in virulence traits in plant pathogens (Singh, Nair & Verma, 2021), which is related to the role of ROS and cell damage (Hernández-Oñate et al., 2012). The varying lifestyle of S. schenckii poses the open question of how to cope with the various ROS stress encountered in this dual organism’s lifestyles.

To our surprise, little is known about the inhibition of fungal catalases. The canonical catalase inhibitors are sodium azide, hydroxylamine, potassium cyanide, salicylic acid (also a molecule involved in plant defense systems), metal ions, and 3-amino-1,2,3-triazol, but no quantitative or structural studies have been carried out with catalases from fungi. The best examples are either mammalian or bacterial purified enzymes (Ma, Deng & Chen, 2017).

One concerning setting is the activation of catalases; one study showed that metformin, a common anti-diabetic drug, activates catalase in a mouse model with tetrachloride-induced severe oxidative liver injury (Dai et al., 2014); thus, the detailed role of catalases in pathogenic fungi could lead to preventive actions in patients undergoing metformin treatment. Additional evidence of catalase activation is the role of the alkaloid piperine in enhancing its activity (Caceres et al., 2017). Another interesting catalase activator is vanillin and vanillic acid in animal models (Salau et al., 2020), suggesting that further research is needed to discover and use antifungal treatments.

The inhibition of catalases may require extensive experimental analysis for each fungal catalase. There are cases where catalases are inhibited with relatively harmless molecules derived from natural products such as tea catechins or plant flavonoids (Pal, Dey & Saha, 2014; Krych & Gebicka, 2013) or simply by ethanol (Temple & Ough, 1975). Another relevant aspect is the inhibition of catalase by natural means, such as targeting heme iron with molecules present in the respiratory burst, such as reactive nitrogen species. Heme binds molecules such as nitric oxide, cyanide, and hydrogen sulfide (Bieza et al., 2015; Milani et al., 2005); thus, exploring another hydrogen peroxide detoxifying enzyme, such as peroxidases, is relevant to the mechanism of invasion and survival of pathogenic fungi of mammalian and plant hosts.

Additionally, a collection of different compounds found in the plant Jacquima macrocara that inhibit the growth and spore germination of Fusarium verticillioides inhibits catalase activity completely at 1.25 mg/mL of the plant extract (Valenzuela-Cota et al., 2019). The repercussions of finding novel antimicrobial compounds that one of its targets is the antioxidant capacity of pathogenic fungi is worth exploring further, not only for human pathogens but also for veterinary purposes and phytopathogenic fungi.

Environmental hazards can also be of interest (Asemoloye, Ahmad & Jonathan, 2018). Asemoloye and colleagues (2018) found that crude oil from an oil spill site at Ugborodo community, Nigeria, induced catalases, laccases, and peroxidases in fungal organisms present in the rhizosphere. These results are relevant for the biodegradation of oil-derived molecules and strong selective pressure for fungi that, as demonstrated, require degrading enzymes such as laccase and an arsenal of antioxidant enzymes but are also strong selective pressure for pathogenic fungi with a free-living stage.

The circadian cycle regulates plant response and ROS production against plant pathogenic fungi (Liang, Dong & Deng, 2022), but question remain on how this mechanism influences other physiological aspects of fungi in the interrelation with plant defense mechanisms that are also regulated by time-of-day manner and ultimately defines the outcome between this interaction. Nevertheless, does this influence the pathogenic state of Sporothrix and other pathogenic fungi, such as Metharizium, in response to light? In particular, survival mechanisms during UV light exposure (Brancini et al., 2022) or the role of conidia formation and other biological aspects of cell differentiation, such as the outcome of light of different wavelengths, have been reported in Metarhizium (Dias et al., 2020). On the other hand, cell differentiation in fungi depends on ROS, specifically for the formation of invasive structures such as the appressorium in Magnaporthe oryzae (Kou, Qiu & Tao, 2019), which is derived from the own metabolism of the fungus via Nox 1 and Nox2 NADPH oxidases (Egan et al., 2007). In vivo measurements of ROS during cell differentiation or invasion could shed light on the role of ROS in dimorphic pathogenic fungi.

All questions regarding the role of catalases and the antioxidant arsenal can be first assessed in alternative infection models, such as the invertebrate insect larvaeTenebrio molitor (Lozoya-Pérez et al., 2021; de Souza et al., 2015). Using T. molitor or Galleria mellonela as a model, relevant information can be obtained from gene knockouts or silencing of catalase genes in saprophytic fungi.

Are other conditions relevant for catalase regulation? Recently, it was found that different species of Sporothrix (S. schenckii, S. brasiliensis, and S. globosa) show lower survival rates due to abnormal cell-wall composition during carbon and nitrogen starvation and are also linked to the virulence phenotype elicited by different members of the Sporothrix complex (Lozoya-Pérez et al., 2020), here catalases and other moonlight antioxidant proteins may be incorrectly linked to the cell wall, making the cells more susceptible to oxidative defenses of the host. One interesting feature to explore is whether catalases and other antioxidant enzymes are downregulated during starvation, which may also reduce virulence.

Finally, do pathogenic fungi possess bifunctional catalases, which may be involved in the free-living lifestyle and have a pivotal role in host invasion? One such example is the bifunctional catalase MkatG1 in the locust-specific pathogen Metarhizium acridum (Li et al., 2017). In this insect pathogen, catalase is induced during exposure to the cuticle and during the formation of the appressorium. In the mutant lacking this catalase, germination and appressorium formation are reduced on locust wings as well as quinone/phenolic compounds production, showing the relevance of this catalase/peroxidase enzyme in host invasion.

Overall, catalases offer the opportunity to revisit their role and can provide potential solutions for antifungal therapies. Linares and colleagues found that anticancer drugs enhance the activity of catalases in C. albicans, which could explain the concomitant infections of this pathogen among patients undergoing chemotherapy (Linares et al., 2006). This suggests that complementary therapies that inhibit the antioxidant arsenal of C. albicans may reduce the complications found during the course of chemotherapy.

The case of the Candida genus is particularly relevant to the study of catalases and pathogenesis. Firstly, members of the Candida genus contain differences in their cell wall components, resulting in differential recognition by the immune system (Navarro-Arias et al., 2019). Secondly, this genus shows a geographic-dependent prevalence and, thus, different phenotypes related to antifungal drugs and virulence determinant production (Ziccardi et al., 2015), rendering it a hot topic to analyze with other aspects such as catalase production. Finally, the relationship of some members of the Candida genus and higher expression levels of virulence factors, resistance to polyenes, azoles, and echinocandins, along with higher catalase expression, is part of the pathogenesis, as demonstrated for Candida glabrata (Figueiredo-Carvalho et al., 2015).

In the case of fungal pathogens, there are still several basic physiological aspects to be explored to fully assess ways of controlling fungal infections and reducing resistance to pharmaceutical treatment. Also, the study of clinical or specific geographical isolates will help to determine virulence and resistance to antifungal drugs (Ziccardi et al., 2015), which may be favored by higher catalase expression or diversity.

Conclusions

Overall, the Sporothrix genus is a neglected disease frequently found in tropical and subtropical areas, with research focused on cell structures such as the cell wall. Here, we propose that other key enzymes related to oxidative stress resistance, specifically catalases, may be target for treatment due to its sequence and unique structural features. With the increasing threat that global warming is posing to all forms of life in the planet, infectious diseases are taking the central stage (El-Sayed & Kamel, 2020). The main threats are the migration and emergence of pathogens in areas that have not been detected previously. They also trigger the selection of more resistant (and perhaps more virulent) strains (El-Sayed & Kamel, 2020). Among the most important environmental factors are temperature and humidity, leading to stressing conditions that ultimately select the most resistant strains with increasing temperature; this may result in the re-distribution of hosts and pathogens (El-Sayed & Kamel, 2020).

We encourage the scientific community to focus efforts on the research of neglected tropical and subtropical diseases as part of humankind’s effort to reduce the effects of global warming. Also, forgotten key enzymes, such as catalases, play an important role in cell physiology that may result in novel targets for treatment if human pathogenic fungi become a bigger burden than they already are.

Supplemental Information

Supplemental Information 1 Representative articles cited in this manuscript, listed alphabetically

Click here for additional data file.

Supplemental Information 2 Dimer prediction of catalases

PDB files of the dimer prediction of S. schenckii catalases.

Click here for additional data file.

Supplemental Information 3 Raw data

Sequences and PDB files used in this study.

Click here for additional data file.

Additional Information and Declarations

Competing Interests

Author Contributions

Data Availability

Bernardo Franco is an Academic Editor for PeerJ. The authors declare that the research was conducted without any commercial or financial relationships that could be construed as a potential conflict of interest.

Naurú Idalia Vargas-Maya conceived and designed the experiments, analyzed the data, prepared figures and/or tables, authored or reviewed drafts of the article, and approved the final draft.

Vianey Olmedo-Monfil analyzed the data, prepared figures and/or tables, authored or reviewed drafts of the article, and approved the final draft.

Jorge Humberto Ramírez-Prado analyzed the data, prepared figures and/or tables, authored or reviewed drafts of the article, and approved the final draft.

Ruth Reyes-Cortés analyzed the data, authored or reviewed drafts of the article, and approved the final draft.

Felipe Padilla-Vaca conceived and designed the experiments, analyzed the data, prepared figures and/or tables, authored or reviewed drafts of the article, and approved the final draft.

Bernardo Franco conceived and designed the experiments, analyzed the data, prepared figures and/or tables, authored or reviewed drafts of the article, and approved the final draft.

The following information was supplied regarding data availability:

All the models and sequences used in this work are available in the Supplementary Files.

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
