# Peer review of "Catalases in the pathogenesis of Sporothrix schenckii research"

_PeerJ, doi:10.7717/peerj.14478_

## Round 0.1 · original submission · Major Revisions

Please follow the suggestions of the reviewers and modify/revise the manuscript at the earliest.

Reviewer 1 ·

Basic reporting

no comment.

Experimental design

no comment.

Validity of the findings

This is a well-written and thoroughly analyzed review article. The authors raised their question in the introduction, 'why have catalases been neglected in pathogenic fungi research as both potential targets for treatment and diagnosis?' However, after reading the manuscript it is still unclear to me why, if catalase is such an important protein to virulence as the authors acclaimed.

Catalase can be an oligomeric enzyme as a number of structures showed in the past. The authors need to analyze or discuss whether this may be true for Sporothrix. Perhaps, AlphaFold-multimer will provide more information than AlphaFold2, which the authors used in this current version of manuscript.

Fig. 2B should be separate to Fig. 2B and Fig. 2C, as the colors don't match the legends.

·

Basic reporting

I consider that the article is sufficiently well written in English, but that it could improve its quality by refining the wording of some sentences. The sentences are sometimes very concrete and I would think that a little more context could be provided so that the reader can understand the underlying idea. In the attached document, I place some comments that address this aspect.
The authors' work delves into catalases, but this is only part of the antioxidant response of Sporothrix schenckii. It would be very worthwhile if they could include the experimental observations found by other authors, such as Ortega et al in 2015, where they compare the response to oxidative stress of S. schenckii and S. brasiliensis, and where they provide us with the routes of signaling possibly involved in the detection and response to oxidative stress in these microorganisms. I suggest including in your analysis the following references:
• Ivy Ortega, Maria Sueli Soares Felipe, Ana Tereza Ribeiro Vasconcelos, Leila Maria Lopes Bezerra, Alessandra Da Silva Dantas, Peroxide sensing and signaling in the Sporothrix schenckii complex: an in silico analysis to uncover putative mechanisms regulating the Hog1 and AP-1 like signaling pathways, Medical Mycology, Volume 53, Issue 1, January 2015, Pages 51–59, https://doi.org/10.1093/mmy/myu069
• Verônica S.P. Castro, Aleksandro S. Da Silva, Gustavo R. Thomé, Patrícia Wolkmer, Jorge L.C. Castro, Márcio M. Costa, Dominguita L. Graça, Daniele C. Oliveira, Sydney H. Alves, Maria R.C. Schetinger, Sonia T.A. Lopes, Lenita M. Stefani, Maria I. Azevedo, Matheus D. Baldissera, Cinthia M. Andrade, Oxidative stress in rats experimentally infected by Sporothrix schenckii, Microbial Pathogenesis, Volume 107, 2017,Pages 1-5, ISSN 0882-4010, https://doi.org/10.1016/j.micpath.2017.03.001.
The quality of the figures can be embellished, in the attached document, I include comments that, if heeded, would help improve the understanding of what is displayed in the figures. In general I have observations regarding the order of the panels, I suggest increasing the size of the font and the elimination of some pieces of information in section 1 B. I suggest the inclusion of a rectangle in figure 1 C and the inclusion of an external group or organism to the phylogenetic tree in Figure 1B. I suggest that the names of the organisms be carefully italicized and an additional figure consisting of another catalase phylogenetic tree addressing the clades mentioned on lines 62-29 be placed present before figure 1. For figure 2, I present comments to standardize the presentation of the structures so that they can be compared with each other more easily. I suggest exchanging the last overlay structure to match what is described in the figure caption. I thank the authors for including the raw files for the analysis of their protein structures and the sequences used for the construction of the phylogenetic trees.
Regarding the scope of the review, I believe that it does meet the editorial criteria to be included in PeerJ, however, the comments of the peer reviewers should be addressed and their and my questions answered throughout the document.
It seems to me that the central idea of placing the oxidative stress response in a dimorphic fungal pathogen is relevant to understanding how environmental conditions evolutionarily drove the jump of Sporothrix from its ecological niche to the human host. It is very interesting, however, I think that the inclusion of other references is necessary to finish landing and strengthen the idea of the authors.

Experimental design

Similarly, I believe that the authors have followed a rigorous methodology for collecting the information that supports their review, although as I mentioned before, the inclusion of other references, search terms or keywords is needed to expand or complement the manuscript.
One of the areas of opportunity that I found is precisely the methodological one. They should be more detailed in the route of the procedures that they followed for the construction of the phylogenetic trees, which method did they choose? Neighbor-joining, maximum parsimony or Bayesian inference? They would do very well to make these details transparent and mention the parameters chosen for its construction. Similarly, they should mention the methodology and parameters used in the other bioinformatics tools: alignment viewer, weblogo, alphafold and mTM-Align. It is necessary that this information be attached to section 2 and not only appear briefly described in the footnotes of the figures. Unfortunately, as they are currently written, they could not be reproduced by other researchers. In relation to the above, it seems to me necessary that the citations to these bioinformatic tools are also found in section 2 as well as in the footnotes of the figures.
As for the organization of the review, it seems to me that part 3 could be divided into categories. In the first section, you could focus on the phylogenetic comparison of the structure of catalases and later on the differences in their expression in the filamentous and yeast forms of S. schenkii. Subsequently, it could be concluded with the importance or contribution that catalases have during the infection of the host in other dimorphic fungi. Additionally, the relevance of catalases in the saprophytic lifestyle of fungi with mycelial phase could be further explored.

Validity of the findings

In this section, I reiterate that the methodology must be explained in more detail so that another researcher can reproduce the comparisons made by the authors.
I think that section 3 of the review is very complete, but that section 1, the introduction, could focus more on S. schenckii and the species of the complex, in such a way that the importance of this fungal pathogen is highlighted.
The argument of reviewing the oxidative stress response of S. schenckii and focusing on catalases is interesting and well-justified, but I believe that further literature review is required to solidly support the review.
Finally, it is required that the work concludes with at least some lines that emphasize the importance of the antioxidant response of S. schenkii, so that future research is aimed at better understanding this fungus that causes sporotrichosis.

Additional comments

Please, attend the recommendations attached in the PDF file. Thank you.

Reviewer 3 ·

Basic reporting

In this study, they proposes a closer look on Sporothrix schenkii as an example of an emerging fungal pathogen with an evolutionary well-adapted saprophytic lifestyle. This manuscript is interesting, but it is not clear whether it is an original article or a review. The literature on Catalases in the pathogenesis of fungus is scarce.

Experimental design

Major comments

Methodology
They could explain how they searched the literature and how they selected studies. Were there selection criteria?
There are not enough details about the pathogenesis of Sporothrix

Figures 1 and 2 show interesting results. I suggest considering it as an Original article, including the Results and Discussion section.

Include a summary table with included studies and their findings.

I suggest including a section on limitations, given the scant literature on Catalases in the pathogenesis of S. schenckii.
To compare the literature on catalases in the pathogenesis of S. schenckii, and other Sporothrix species such as S. brasiliensis and globosa.

Validity of the findings

The review is very brief. I suggest giving more details about catalases in the role of Sporothrix pathogenesis

I suggest including a conclusion section

Additional comments

Review the title "Catalases in the pathogenesis of Sporothrix schenckii research"

---

## Round 0.2 · accepted · Accept

Your paper is accepted for publication.

Reviewer 1 ·

Basic reporting

no comment

Experimental design

no comment

Validity of the findings

no comment

·

Basic reporting

Thank you very much for listening to the comments and answering the questions raised by me and by the rest of the reviewers. Likewise, I thank the authors for including the suggested references. Similarly, thank you for correcting the figures and addressing the recommendations to improve their quality and also for including figure 3.

Experimental design

The methodology is now more detailed and I now find it sufficient to be reproduced by other researchers.

Validity of the findings

Thank you very much for including the conclusions section.

Additional comments

Thank you very much for improving the manuscript. I attach only a few small observations in the following word document.